# Analysis of the Quality of SLR Station Coordinates Determined from Laser Ranging to the LARES Satellite

**DOI:** 10.3390/s21030737

**Published:** 2021-01-23

**Authors:** Stanisław Schillak, Paweł Lejba, Piotr Michałek

**Affiliations:** Centrum Badań Kosmicznych Polskiej Akademii Nauk (CBK PAN), 00-716 Warszawa, Bartycka 18A, Obserwatorium Astrogeodynamiczne CBK PAN, 62-023 Borówiec, Poland; plejba@cbk.poznan.pl (P.L.); misiek@cbk.poznan.pl (P.M.)

**Keywords:** satellite geodesy, satellite laser ranging (SLR), reference frames, station coordinates determination, law-altitude satellite orbits, LARES and LAGEOS satellites

## Abstract

The LARES (LAser RElativity Satellite) was built by the Italian Space Agency (ASI) and launched on 13 February 2012 by the European Space Agency. It is intended for studying the Lense–Thirring effect resulting from general relativity as well as for geodynamic studies and satellite geodesy. The satellite is observed by most ground laser stations. The task of this work is to determine the station coordinates and to assess the quality of their determination by comparison with the results from the LAGEOS-1 and LAGEOS-2 satellites. Observation results in the form of normal points (396,105 normal points in total) were downloaded from the EUROLAS Data Center for the period from 29 February 2012 to 31 December 2015. Seven-day orbital arcs were computed by the NASA GSFC GEODYN-II software, determining the coordinates of seventeen selected measuring stations. The average Root Mean Square (RMS) (15.1 mm) of the determined orbits is nearly the same as for LAGEOS (15.2 mm). The stability of the coordinates of each station (3DRMS) is from 9 mm to 46 mm (for LAGEOS, from 5 mm to 15 mm) with the uncertainty of determining the coordinates of 3–11 mm (LAGEOS 2–7 mm). The combined positioning for the LARES + LAGEOS-1 + LAGEOS-2 satellites allows for the stability of 5–18 mm with an uncertainty of 2–6 mm. For most stations, this solution is slightly better than the LAGEOS-only one.

## 1. Introduction

The LARES (LAser RElativity Satellite) is mainly intended to study the Lense–Thirring effect resulting from general relativity [1,2,3]. This effect arises when a rotating massive body with a large moment of inertia drags a locally defined inertial system in its gravitational field (frame dragging). Frame dragging should exist regardless of the primary moment of inertia value (provided it is non zero). The results of this research were presented in many papers, e.g., [4,5,6]. Several papers describe the determination of tidal effects from the LARES [7,8]. The LARES can also be used in satellite geodesy [9]. The problem of including low-altitude geodesy satellites in determining the International Terrestrial Reference Frame (ITRF) has been a topic of discussion for many years [10]. The International Laser Ranging Service (ILRS) [11] is considering incorporating the LARES satellite into the ITRF development. However, there is a lack of articles that present research on these actions. The work to date containing the results of laser observations of this satellite concerns its basic applications in studying the Lense–Thirring effect. The presented work is intended to answer several important questions regarding the possible inclusion of the LARES in ITRF development. The basic questions to be answered by the work are: what is the quality of the coordinates of the stations determined from the laser measurements of the LARES, what is the difference in the results from determining the coordinates of the stations from the LAGEOS-1 and LAGEOS-2, what is the quality of results from the combined solution LARES + LAGEOS, and can this solution be used for ITRF determination?

Why is the LARES the most useful low-altitude geodetic satellite to co-establish the ITRF? Among the geodetic satellites, the basis for determining the coordinates of the stations are the LAGEOS-1 and LAGEOS-2. Etalon-1 and Etalon-2 satellites are also used for this purpose, but due to their high orbit (19,000 km) there are too few laser measurements to improve the quality of the coordinates determined from LAGEOS. The list of current passive geodetic satellites, which are in the shape of a sphere for more accurate determination of the center of mass of the satellite, is presented in Table 1.

Among the satellites presented in Table 1, Ajisai’s diameter is too large, causing significant increase in non-gravitational effects. Starlette, Stella [12], and Larets are too low, which causes additional drag from the atmosphere and the need to use high Earth’s gravitational field coefficients, while LARES is most favorable for its use as the third satellite after the LAGEOS satellites to determine the position of stations. Hence its choice in this work. The LARES satellite is pictured in Figure 1.

The LARES satellite, built by the Italian Space Agency (ASI), was launched on 13 February 2012 from the ESA Kourou Center in French Guiana using the Vega VV01 rocket. The satellite is moving in a circular orbit at a distance of 1450 km with an inclination of 69.5 degrees over a period of 1.9 h. The satellite is only intended for laser observations. It is a sphere with a diameter of 36.4 cm and a mass of 386.8 kg (the densest object in the solar system). On the surface it has 92 corner cube retroreflectors with a diameter of 38.1 mm and a depth of 27.9 mm each. The satellite is observed by most ground laser stations.

The main advantages of LARES satellite for its use in satellite geodesy are as follows: its spherical shape, very well-defined center of mass, high mass (M) (low influence of non-gravitational effects), small cross section (A) (low influence of non-gravitational effects), small cross-section to mass, the ratio A/M = 2.69 × 10^−4^ m^2^/ kg (the densest known object in the solar system (15.3 g/cm^3^)), its circular orbit (an eccentricity of 0.0008). 

Its disadvantage is its low satellite orbit (1450 km), which entails a significant impact due to high harmonics of the Earth’s gravity field (Earth’s gravity field coefficients up to 100 × 100), earth albedo, and residual atmospheric drag [5].

## 2. Determination of the SLR Station Coordinates

The orbit of the LARES and SLR station coordinates have been determined using the NASA GSFC GEODYN-II orbital software [13] from the laser ranging data of 17 selected stations for the period from 29 February 2012 to 31 December 2015. SLR results were taken from the EUROLAS Data Center (EDC) in the form of 30-s normal points [14]. Weekly arcs were used in the computations, and a total of 200 orbital arcs were computed from 396,105 normal points. The average RMS of determined orbits over the entire almost four-year period was 15.1 mm. For the same orbital arcs computed for the LAGEOS-1 + LAGEOS-2, the average RMS was nearly the same (15.2 mm). The models and parameters used for the LARES in the GEODYN-II orbital software are given in Table 2.

From among the force models and parameters presented in Table 2, additional tests were carried out to determine the optimal coefficients of the gravitational field, the time-variable gravity, the atmospheric drag, empirical accelerations, and the corrections of satellite center-of-mass (COM) for each station. The control parameters were the orbital RMS, 3DRMS of the station coordinates, and the standard deviation of the coordinate determination. All tests were performed for Yarragadee results. The most favorable model of the gravitational field is for coefficients up to degree and order 100 × 100; above these values, the control parameters do not decrease and are constant. The influence of the time-variable of the gravitational field is imperceptible. The influence of atmospheric drag was not found. On the other hand, the selection of empirical accelerations is important. The best results for the LARES was obtained for all three components applied: along-track, cross-track, and radial for constant and once-per-revolution accelerations in 14 sets per weekly solution. The value of the solar radiation pressure coefficient from 1.07 to 1.13 was also checked. No significant differences in the results were found. The study did not take into account the COM corrections for individual stations [25], and a constant value of 13.1 cm was adopted. The aim of the work is to assess the quality of the determined station coordinates; hence the determination of the full Range Bias (RB) (without the RB part in COM) for each station is important [26]. For the same reason, RB was also not included in the results.

Due to the non-convergence of the iterative process, 21 arcs were rejected, mainly due to the low number of normal points (in the most cases below 900 per arc), especially during the winter time. The list of all SLR stations that were observing the LARES in the period from 29 February 2012 to 31 December 2015 is given in Table 3.

To obtain the most accurate results, three scenarios for the selection of core stations were tested, the results of which formed the orbit of the LARES. The tests were performed using the parameters and models given in Table 2 for all 40 stations, which in the period from 29 February 2012 to 31 December 2015 were observing the LARES.

Scenario 1: 15 SLR stations selected on the basis of the quality assessment and the number of normal points of the LAGEOS were used.

Scenario 2: all 40 stations that observed the LARES during the period were used.

Scenario 3: based on the results of scenario 2, the 17 best stations were selected as the core stations for creating the orbit of the LARES.

The evaluation of the obtained results was carried out for each station based on the number of normal points, the average 3DRMS of coordinates determined, and the continuity of the observations. The best results were achieved for scenario 3 and these 17 stations were used as the core stations for further analysis (Table 3, bold).

## 3. Results and Discussion

The geocentric coordinates of the stations computed by the NASA GSFC GEODYN-II software were processed to the topocentric coordinates North, East, and Up by means of the Borkowski transformation [27]. The results include differences from SLRF2014 [22]. From these results, the average RMS was determined for each component, which in 3D characterizes the accuracy of the coordinates determined [12]. The precision of results in the form of the standard deviation of determined coordinates was also computed (Table 4). An important indicator for assessing the quantity and quality of observations for each station (percent of accepted arcs) is presented in Table 4. It points out that this percentage is slightly lower for the LARES (LA), while it is higher for most of the three satellites (LA + L12). The stations are listed in Table 4 according to their continental location. Higher quality of results for European stations is clearly visible, and is caused by a large number of stations in a small area (8 stations) and therefore a large number of measurements (this orbit segment is better determined than for other continents). It should be emphasized that the quality of observations of the stations placed on other continents is not connected with this effect, which is confirmed by the LAGEOS results for these stations (Table 4).

To compare the results of the LARES with the LAGEOS-1 and LAGEOS-2 satellites, the same weekly arcs for LAGEOS from the same stations were computed using the GEODYN-II software. Parameters and models of the orbital software are provided in [28]. The results are shown in Table 4 (L12). Higher quality of coordinates determined from the LAGEOS is clearly visible. In the case of the LAGEOS, we don′t observe dependence of the quality on the continental location as in the case of LARES.

The number of normal points for the three satellites (LARES, LAGEOS-1, LAGEOS-2) for each station on the whole period of this study is presented on Figure 2. This number for each station is similar for all satellites. In addition, orbital mean RMS of normal points of each station for all satellites is on the same level (Figure 3) with an average of 15.2 mm (LARES) and 15.1 mm (LAGEOS1 + LAGEOS2) for all points of the 17 stations. An important parameter that indicates the stability of the station’s range bias over several years is long-term stability. The results for each station and satellite are presented in Figure 4 and Table 5. Much better long-term stability for the LARES is visible.

Examples of the determined North, East, and Up components for the LARES and LAGEOS satellites for the Zimmerwald and Yarragadee stations are shown in Figure 5 and Figure 6, respectively. The average deviation from SLRF2014 and the stability of the determined position (3DRMS) are given for each component.

The most important element for the future inclusion of the LARES satellite in ITRF is the determination of the station coordinates from normal equations for all three satellites combined (LARES + LAGEOS-1 + LAGEOS-2).

The best parameter that determines the quality of the station coordinates is the average 3DRMS of the three components (station, position, and stability) (Equations (1) and (2)):(1)$$RMS_{X}=\sqrt{\frac{\sum_{i=1}^{n}{\left(X_{i}-\bar{X}\right)}^{2}}{n-1}}$$
where ***i*** is the arc number, ***X*** is value of component X for each arc, $\bar{X}$ is the mean value of the components $X_{i}$, and ***n*** is the number of arcs. The RMS of the components ***Y*** and ***Z*** are computed analogously. The total RMS for all components (total station position stability) is computed from the Formula (2):(2)$$3DRMS=\sqrt{\frac{RMS_{X}^{2}+RMS_{Y}^{2}+RMS_{Z}^{2}}{3}}$$
the components North, East, and Up are computed analogously (instead of ***X, Y, Z***) and the final result (2) has to be the same.

The 3DRMS for each station and the three solutions is presented in Table 4 and Figure 7. The least accurate values were obtained for the LARES; much more accurate results for determining the position from this satellite for the European stations are clearly visible (7810, 7839, 7840, 7841, 7941, 8834). They are at the level of slightly weaker stations for LAGEOS. In contrast to the solution from all three satellites, as many as 10 stations have better results than the solution from LAGEOS alone, which indicates that the LARES + LAGEOS-1 + LAGEOS-2 solution after further research and analysis can be successfully used in the development of ITRF.

The precision of coordinate determination is defined by 3D standard deviation (*σ*) as follows (3):(3)$$\sigma =\sqrt{\frac{{\bar{\sigma }}_{X}^{2}+{\bar{\sigma }}_{Y}^{2}+{\bar{\sigma }}_{Z}^{2}}{3}}$$
where $\bar{\sigma }$*_X_*, $\bar{\sigma }$*_Y_*, $\bar{\sigma }$*_Z_* are average standard deviations of the determined components *X*, *Y*, *Z*. The results presented in Table 4 and Figure 8 show a much better precision for a combined solution from three satellites for almost all stations, except Svetloe (1888), which confirms the previous conclusion that the solution for three satellites gives better results for coordinate determination. In the process of the standard deviation of the station coordinate determination, two parameters are crucial. First is the number of normal points per arc and second is the orbital RMS of these points. We have to bring into account also satellite trajectory. Both LAGEOS 1 and 2 have a significantly higher number of normal points (one and half times more than LARES), and the orbital RMS is the same as LARES (Figure 3), but the LAGEOS satellites have significantly better coverage of observations than LARES does. Therefore, the LAGEOS + LARES data can give better results than the LAGEOS data alone, mainly because it features a much higher number of normal points.

Examples of the determined North, East, and Up components for each arc for LAGEOS and LARES + LAGEOS for the Zimmerwald and Yarragadee stations are presented in Figure 9 and Figure 10, respectively.

The presented results show good agreement for both solutions (L12 and L12 + LA) with an indication of small improvement for the solution from three satellites. For Yarragadee station (Figure 10), both solutions for all three components show an annual wave with amplitude of about 5 mm. For each station, a comparison of average Range Bias and orbital RMS was performed for all three satellites: LARES, LAGEOS-1, and LAGEOS-2 (Table 5). It is worth emphasizing that high consistency of results for all three satellites for each station has been achieved.

## 4. Conclusions

We assessed the quality of station coordinates determined from the results of laser observations of the LARES satellite, noting that they are much lower than in the case of LAGEOS satellites. The station coordinates obtained from LARES are additionally dependent on the continental location of the station. Positive results were obtained for the results from combined solution of data from the LARES, LAGEOS-1, and LAGEOS-2. The results are even slightly better than those from LAGEOS. Very good agreement between average Range Bias and orbital RMS was found for all three satellites. The average orbital RMS for the LARES is ±15.1 mm for all stations and is nearly the same as for the LAGEOS.

The accuracy of the station coordinates, defined as the average 3DRMS position for the LARES, changed from 9.5 mm to 45.9 mm, while for the LAGEOS satellites it changed from 5.2 mm to 15.4 mm, similar to the combined solution from three satellites (LARES, LAGEOS-1, LAGEOS-2) from 5.2 mm to 18.3 mm (Table 4). The station position accuracy for the LARES satellite is much better for European stations than for other stations. This is due to the large number of observations in Europe, as 8 of 17 stations carried out observations in Europe. On other continents, this number is exceedingly small (1–3 stations).

Position uncertainty, defined as the average 3D standard deviation of the coordinate determination for individual stations for LARES, is in the range of 3.4 mm to 11.1 mm. For LAGEOS, it is much better at 1.9 mm to 6.7 mm. For LARES + LAGEOS, the best result, it is from 1.7 mm to 5.7 mm.

An important problem with the LARES observations is the lack of normal points in some weeks, especially in winter time, which causes the need to remove arcs due to the non-convergence iterative process. For some of the 40 stations there are too few normal points per week, which results in rejection of arcs due to the need to meet the criteria of quantity (>20 normal points per arc) and quality (<average 3 sigma). The result is that only 20% of all normal points for some stations will be left.

Will the results from many geodetic passive satellites (LAGEOS-1, LAGEOS-2, LARES, Etalon-1, Etalon-2, Ajisai, Starlette, Stella, Larets) [26] allow for better accuracy in determining the station coordinates? A larger number of normal points for each station can improve the results obtained only from LAGEOS satellites. This requires further studies to answer this question; it is an extremely important matter for the creation of subsequent International Terrestrial Reference Frames (ITRF) for release.

In the near future, there is a plan to launch the LARES 2 into an orbit similar to the LAGEOS satellites (altitude 5890 km) [29], which will remove the effect of atmospheric drag, significantly reduce the number of Earth’s gravity field coefficients, and also significantly reduce the influence of the Earth’s albedo. This should ensure the quality of position determination at the level of the LAGEOS, and in the overall solution, due to the increase in the number of normal points, increase the accuracy of stations coordinates determination.

The combined solution from three satellites (LARES + LAGEOS-1 + LAGEOS-2) is better than the LAGEOS-only option. It is a good signal for including the LARES satellite and, in the future, LARES-2 to ITRF also, but this work needs additional activity of ILRS analysis centers.

## Figures and Tables

**Figure 1 sensors-21-00737-f001:**
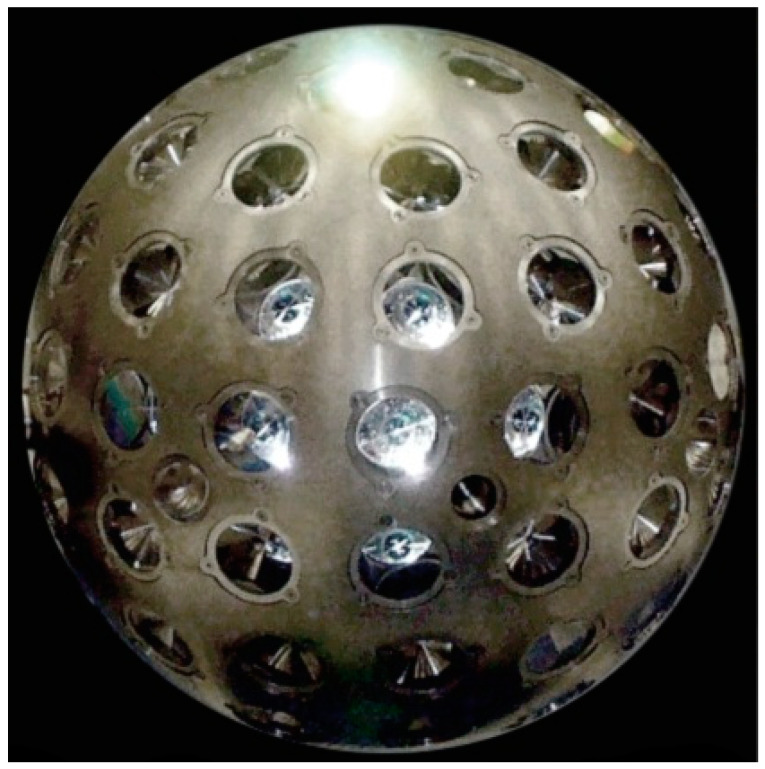
LAser RElativity Satellite (LARES). http://www.lares-mission.com/foto.asp.

**Figure 2 sensors-21-00737-f002:**
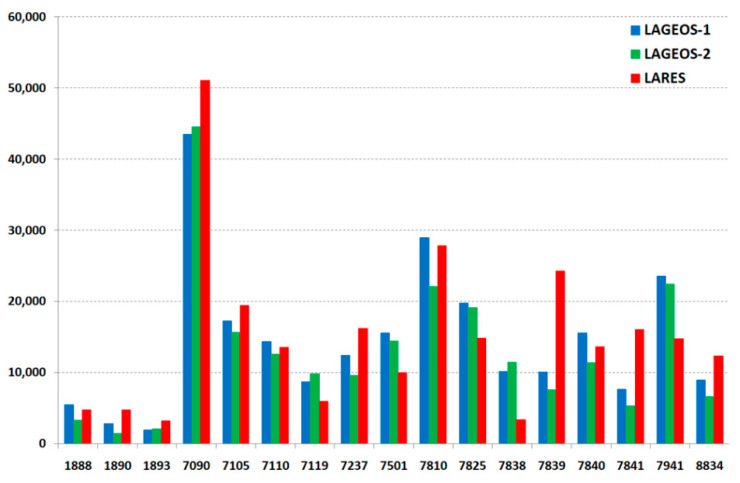
The number of normal points per station in the period from 29 February 2012 to 31 December 2015.

**Figure 3 sensors-21-00737-f003:**
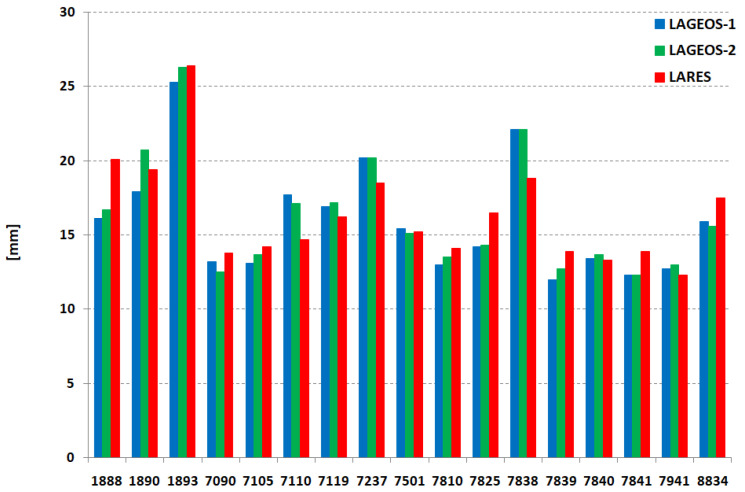
The average orbital RMS per station.

**Figure 4 sensors-21-00737-f004:**
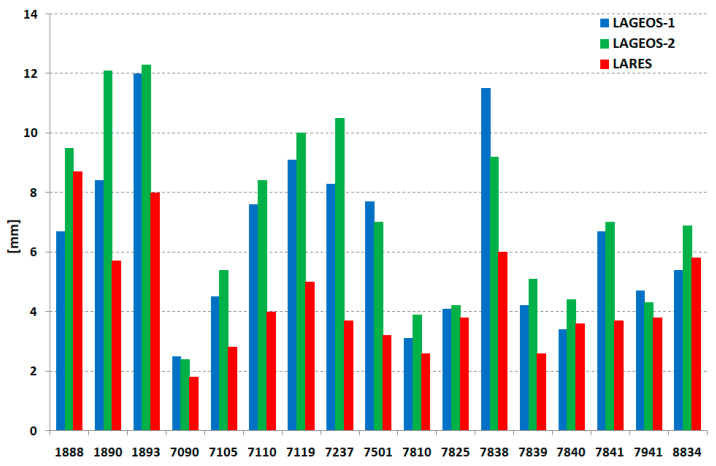
Long-term stability per station.

**Figure 5 sensors-21-00737-f005:**
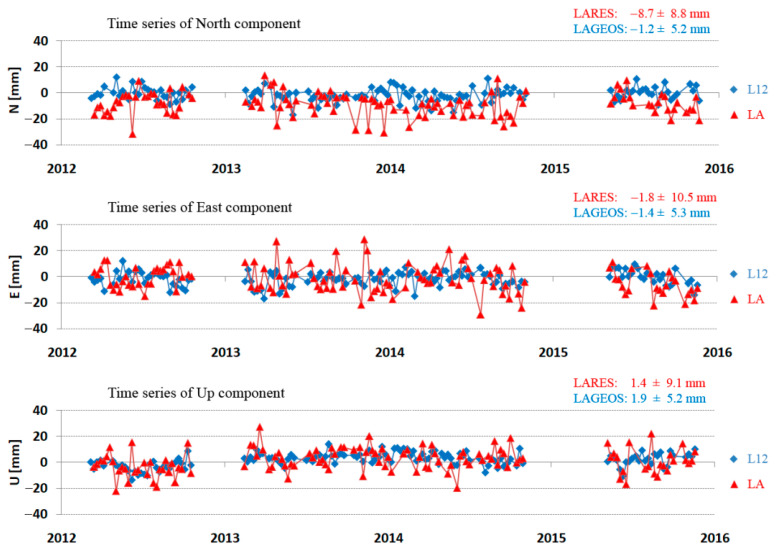
Time series of the North, East, and Up components for Zimmerwald station (7810) for LARES (red) and LAGEOS (blue) in the period from 2012 to 2015.

**Figure 6 sensors-21-00737-f006:**
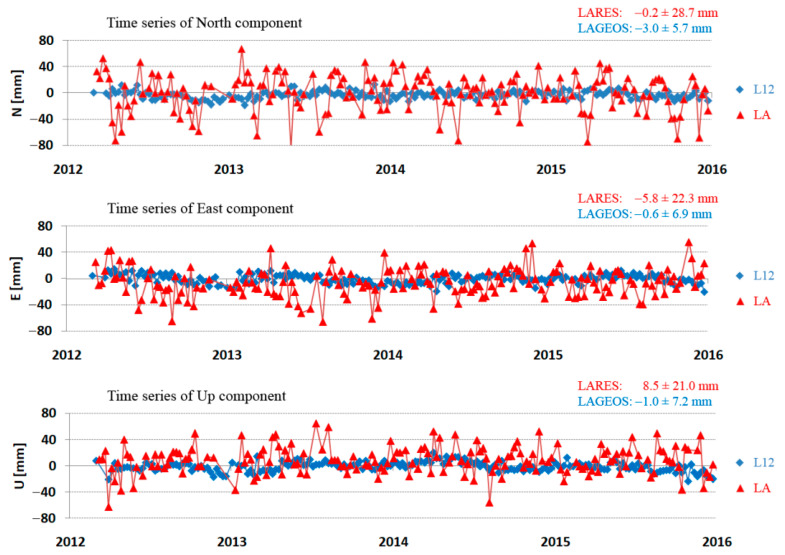
Time series of the North, East, and Up components for Yarragadee station (7090) for LARES (red) and LAGEOS (blue) in the period from 2012 to 2015.

**Figure 7 sensors-21-00737-f007:**
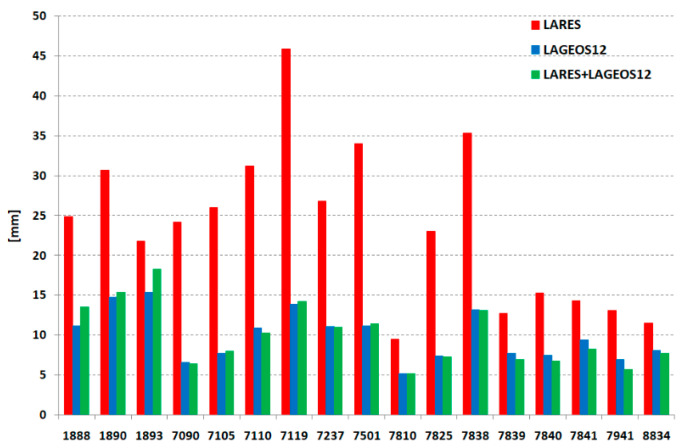
3DRMS for each station for the three solutions: only LARES (red), LAGEOS-1 + LAGEOS-2 (blue), LARES + LAGEOS-1 + LAGEOS-2 (green).

**Figure 8 sensors-21-00737-f008:**
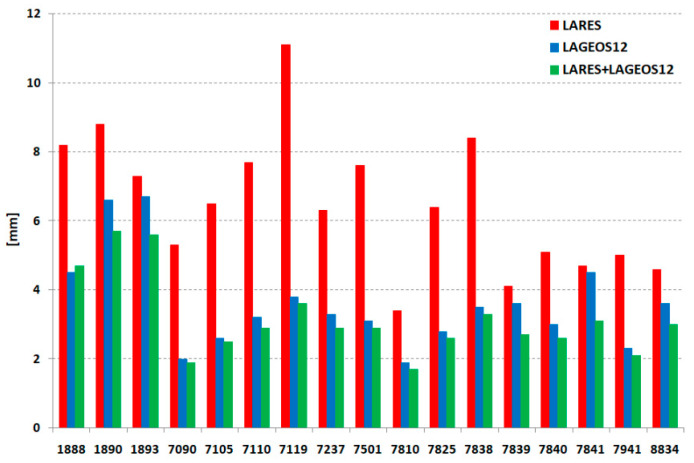
The average 3D precision for each station for three solutions: only LARES (red), LAGEOS-1 + LAGEOS-2 (blue), LARES + LAGEOS-1 + LAGEOS-2 (green).

**Figure 9 sensors-21-00737-f009:**
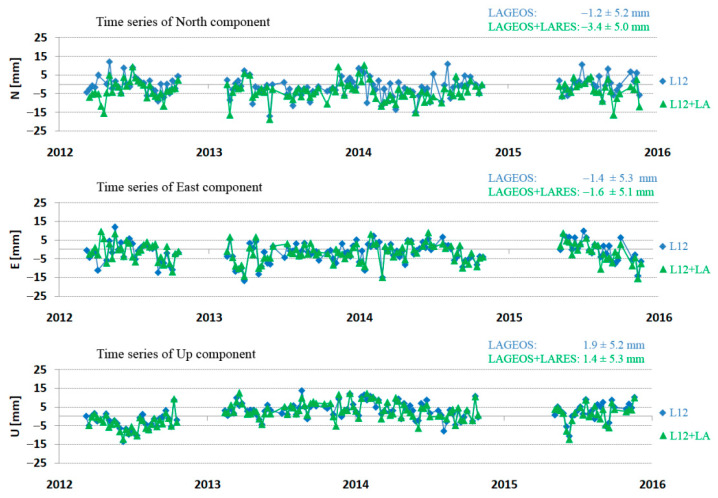
Time series of the North, East, and Up components for Zimmerwald station (7810) for LAGEOS + LARES (green) and LAGEOS (blue) in the period from 2012 to 2015.

**Figure 10 sensors-21-00737-f010:**
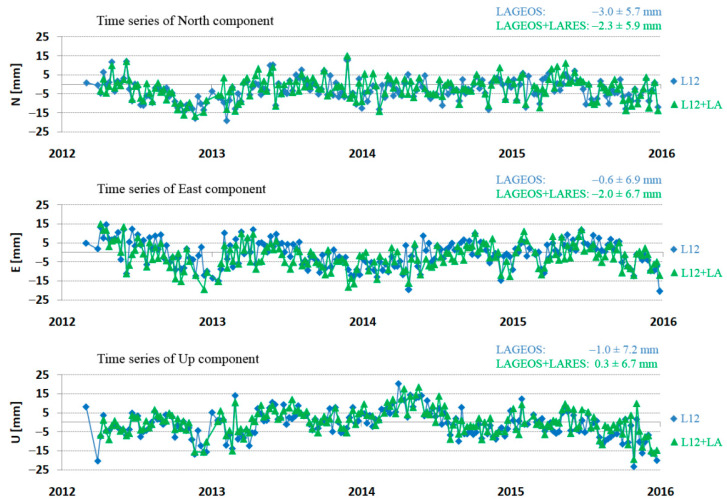
Time series of the North, East, and Up components for Yarragadee station (7090) for LAGEOS + LARES (green) and LAGEOS (blue) in the period from 2012 to 2015.

**Table 1 sensors-21-00737-t001:** Satellite Laser Ranging (SLR) geodetic satellites.

Satellite	Altitude [km]	Diameter [cm]	Inclination [deg]	Density [g/cm^3^]	Launch Year
LAGEOS-1	5850	60	110	3.6	1976
LAGEOS-2	5625	60	53	3.6	1992
Etalon-1	19,105	129	65	1.3	1989
Etalon-2	19,135	129	65	1.3	1989
Ajisai	1485	215	50	0.1	1986
Starlette	812	24	49	6.6	1975
Stella	804	24	99	6.6	1993
Larets	691	24	98	3.2	2003
LARES	1450	36	69	15.3	2012

**Table 2 sensors-21-00737-t002:** Force models and parameters from the GEODYN-II orbital software in the LARES (LAGEOS data, if differing, are given in parentheses).

**Force Models**
Earth gravity field: EGM2008 100 × 100 [15] (20 × 20)
Earth tides: IERS conventions 2003 [16]
Earth tide model: EGM96
Ocean tide model: GOT99.2 [17]
Third-body gravity: Moon, Sun, and planets: DE403 [18]
Solar radiation pressure coefficient: C_R_ = 1.07 (1.13)
Tidal constants k_2_, k_3_, and phase k_2_: 0.3019, 0.093, 0.0 [19]
Earth albedo [13]
Dynamic polar motion [13]
Relativistic corrections [13]
Atmospheric density model: MSIS-86 [20] (not used)
**Constants**
Earth gravity parameter (GM): 3.986004415 × 10^14^ m^3^/s^2^
Speed of light: 299,792.458 km/s
Semi-major axis of the Earth: 6378.13630 km
Inverse of the Earth’s flattening: 298.25642
**Reference Frame**
Inertial reference frame: J2000.0 Coordinate reference system: True of Date defined at 0.0 h of the first day of each arc
Station coordinates and station velocities: SLRF2014 for epoch 2010.0 [21,22]
Precession and nutation: IAU 2000
Polar motion: C04 IERS
Tidal uplift: Love model H2 = 0.6078, L2 = 0.0847 [19]
Pole tide [13]
**Estimated Parameters**
Satellite state vector (6 parameters)
Station geocentric coordinates (3 parameters)
Acceleration parameters along-track, cross-track, and radial (constant and once per revolution) (14 sets per weekly solution) (2 sets per week)
**Measurement Model**
Observations: 30-s normal points from EUROLAS Data Center (120-s)
Laser pulse wavelength: 532 nm
LARES center of mass correction: 13.1 cm (25.1 cm)
LARES cross section area: 0.1041 m^2^ (0.2827 m^2^)
Mass of LARES: 386.8 kg (LAGEOS-1: 406.965 kg, LAGEOS-2: 405.380 kg)
Tropospheric refraction: Mendes–Pavlis model [23,24]
**Editing Criteria**
Residual of normal points >5*σ*
Cut-off elevation <10 degrees
Station coordinates: below 20 normal points per station and arc + >3D sigma of position determination + >3DRMS for each component North, East, Up
**Numerical Integration**
Integration: Cowell’s method
Orbit integration step size: 30 s (120 s)
Arc length: 1 week

**Table 3 sensors-21-00737-t003:** List of all SLR stations which observed the LARES from 29 February 2012 to 31 December 2015. Lines in bold: core stations.

Station Number	SLR Station Name	Number of Weekly Arcs/Accepted Arcs	Number of Accepted Normal Points	Number of Normal Points Per arc	Date of First Arc	Date of Last Arc
					Year-Month-Day	Year-Month-Day
1824	Kiev (Ukraine)	97/32	1218	38	120,321	151,223
1868	Komsomolsk-na-Amure (Russia)	63/17	476	28	120,509	151,223
1873	Simeiz (Ukraine)	94/49	2651	54	120,327	151,223
1874	Mendeleevo (Russia)	9/3	67	22	141,119	150,909
1879	Altay (Russia)	102/42	1676	40	120,425	151,209
1886	Arkhyz (Russia)	88/37	1219	33	120,912	151,223
1887	Baikonur (Kazachstan)	76/33	1771	54	120,502	150,902
**1888**	**Svetloe (Russia)**	**102/66**	**4737**	**72**	**120,307**	**151,223**
1889	Zelenchukskya (Russia)	77/27	976	36	120,404	151,223
**1890**	**Badary (Russia)**	**135/74**	**4779**	**65**	**120,307**	**151,223**
1891	Irkutsk (Russia)	10/6	190	32	150,715	151,223
**1893**	**Katzively (Ukraine)**	**110/58**	**3218**	**55**	**120,307**	**150,916**
7080	McDonald (Texas-USA)	118/35	1175	34	120,321	150,527
**7090**	**Yarragadee (Australia)**	**183/171**	**51,084**	**299**	**120,307**	**151,223**
**7105**	**Greenbelt (Maryland-USA)**	**171/139**	**19,400**	**140**	**120,307**	**151,216**
**7110**	**Monument Peak (California-USA)**	**171/134**	**13,546**	**101**	**120,307**	**151,216**
**7119**	**Haleakala (Hawaii-USA)**	**153/93**	**5958**	**64**	**120,307**	**151,216**
7124	Tahiti (French Polinesia)	107/46	2205	48	120,307	151,209
**7237**	**Changchun (China)**	**180/142**	**16,171**	**114**	**120,307**	**151,223**
7249	Beijing (China)	80/37	1772	48	120,314	150,513
7308	Koganei (Japan)	35/8	367	46	120,307	140,305
7358	Tanegashima (Japan)	3/1	16	16	120,321	120,411
7359	Daedeok (Korea)	40/14	593	42	130,828	141,112
7403	Arequipa (Peru)	153/78	4315	55	120,307	151,223
7405	Concepcion (Chile)	49/18	650	36	120,307	140,319
7406	San Juan (Argentina)	102/58	5118	88	120,307	141,029
**7501**	**Hartebeesthoek (South Africa)**	**145/76**	**10,004**	**132**	**120,307**	**151,223**
**7810**	**Zimmerwald (Switzerland)**	**132/121**	**27,844**	**230**	**120,314**	**151,118**
7820	Kunming (China)	30/17	609	36	130,102	140,521
7821	Shanghai (China)	124/48	1775	37	120,307	151,223
7824	San Fernando (Spain)	79/16	532	33	120,418	151,125
**7825**	**Mount Stromlo (Australia)**	**182/142**	**14,799**	**104**	**120,307**	**151,223**
7827	Wettzell (Germany)	3/2	228	114	151,118	151,209
**7838**	**Simosato (Japan)**	**136/56**	**3409**	**61**	**120,328**	**151,223**
**7839**	**Graz (Austria)**	**169/141**	**24,257**	**172**	**120,307**	**151,223**
**7840**	**Herstmonceux (UK)**	**177/129**	**13,663**	**106**	**120,314**	**151,223**
**7841**	**Potsdam (Germany)**	**159/129**	**16,040**	**124**	**120,307**	**151,223**
7845	Grasse (France)	73/31	1643	53	120,307	140,827
**7941**	**Matera (Italy)**	**171/138**	**14,752**	**107**	**120,307**	**151,223**
**8834**	**Wettzell (Germany)**	**159/117**	**12,306**	**105**	**120,307**	**151,223**

**Table 4 sensors-21-00737-t004:** LARES-results of station positions determination. **LA:** LARES, **L12:** LAGEOS-1 + LAGEOS-2, **LA + L12:** LARES + LAGEOS-1 + LAGEOS-2.

Station Number	SLR Station Name	3DRMS Coordinates [mm]			Standard Deviation of The Station Coordinates Determination [mm]			Percent of Accepted Arcs%		
		LA	L12	LA + L12	LA	L12	LA + L12	LA	L12	LA + L12
**Europe**										
1888	Svetloe (Russia)	24.9	11.2	13.5	8.2	4.5	4.7	65	70	80
1893	Katzively (Ukraine)	21.8	15.4	18.3	7.3	6.7	5.6	53	54	72
7810	Zimmerwald (Switzerland)	9.5	5.2	5.2	3.4	1.9	1.7	92	92	92
7839	Graz (Austria)	12.7	7.7	6.9	4.1	3.6	2.7	83	89	87
7840	Herstmonceux (UK)	15.3	7.5	6.8	5.1	3.0	2.6	73	92	91
7841	Potsdam (Germany)	14.3	9.4	8.3	4.7	4.5	3.1	81	81	84
7941	Matera (Italy)	13.1	6.9	5.7	5.0	2.3	2.1	81	94	91
8834	Wettzell (Germany)	11.5	8.1	7.7	4.6	3.6	3.0	74	71	81
**North America**										
7105	Greenbelt (Maryland-USA)	26.0	7.7	8.0	6.5	2.6	2.5	81	89	90
7110	Monument Peak (California-USA)	31.2	10.9	10.3	7.7	3.2	2.9	78	93	92
**Pacific**										
7119	Haleakala(Hawaii -USA)	45.9	13.9	14.2	11.1	3.8	3.6	61	90	94
**East Asia**										
1890	Badary (Russia)	30.7	14.8	15.4	8.8	6.6	5.7	55	53	72
7237	Changchun (China)	26.8	11.1	11.0	6.3	3.3	2.9	79	87	89
7838	Simosato (Japan)	35.3	13.2	13.1	8.4	3.5	3.3	41	87	88
**Australia**										
7090	Yarragadee (Australia)	24.2	6.6	6.4	5.3	2.0	1.9	93	97	96
7825	Mount Stromlo (Australia)	23.0	7.4	7.3	6.4	2.8	2.6	78	92	93
**Africa**										
7501	Hartebeesthoek (RPA)	34.0	11.2	11.4	7.6	3.1	2.9	52	87	85

**Table 5 sensors-21-00737-t005:** The average Range Bias, long-term stability, and orbital RMS for SLR stations.

	LAGEOS-1			LAGEOS-2			LARES		
Station Number	Number of Normal Points	Range Bias and Long Term Stability [mm]	RMS [mm]	Number of Normal Points	Range Bias and Long Term Stability [mm]	RMS [mm]	Number of Normal Points	Range Bias and Long Term Stability [mm]	RMS [mm]
1888	5477	1.0 ± 6.7	16.1	3299	0.7 ± 9.5	16.7	4737	0.4 ± 8.7	20.1
1890	2841	2.8 ± 8.4	17.9	1461	−1.4 ± 12.1	20.7	4779	−1.2 ± 5.7	19.4
1893	1974	−0.9 ± 12.0	25.3	2071	−0.8 ± 12.3	26.3	3218	−11.1 ± 8.0	26.4
7090	43,479	0.2 ± 2.5	13.2	44,535	−0.2 ± 2.4	12.5	51,084	1.5 ± 1.8	13.8
7105	17,244	−1.0 ± 4.5	13.1	15,658	−0.4 ± 5.4	13.7	19,400	1.2 ± 2.8	14.2
7110	14,338	1.7 ± 7.6	17.7	12,596	5.6 ± 8.4	17.1	13,546	3.5 ± 4.0	14.7
7119	8703	2.1 ± 9.1	16.9	9856	2.9 ± 10.0	17.2	5958	1.7 ± 5.0	16.2
7237	12,404	1.2 ± 8.3	20.2	9567	0.4 ± 10.5	20.2	16,171	1.7 ± 3.7	18.5
7501	15,533	0.5 ± 7.7	15.4	14,417	2.7 ± 7.0	15.1	10,004	2.3 ± 3.2	15.2
7810	28,923	−0.1 ± 3.1	13.0	22,073	−0.6 ± 3.9	13.5	27,844	2.9 ± 2.6	14.1
7825	19,727	−1.9 ± 4.1	14.2	19,133	−1.6 ± 4.2	14.3	14,799	−2.0 ± 3.8	16.5
7838	10,129	1.2 ± 11.5	22.1	11,436	1.8 ± 9.2	22.1	3409	0.5 ± 6.0	18.8
7839	10,103	−0.5 ± 4.2	12.0	7612	−1.7 ± 5.1	12.7	24,257	3.0 ± 2.6	13.9
7840	15,554	4.2 ± 3.4	13.4	11,399	4.2 ± 4.4	13.7	13,663	1.1 ± 3.6	13.3
7841	7633	−0.5 ± 6.7	12.3	5332	−1.2 ± 7.0	12.3	16,040	0.3 ± 3.7	13.9
7941	23,552	−2.0 ± 4.7	12.7	22,418	−2.3 ± 4.3	13.0	14,752	0.2 ± 3.8	12.3
8834	8960	−9.6 ± 5.4	15.9	6621	−9.2 ± 6.9	15.6	12,306	−9.2 ± 5.8	17.5

## Data Availability

Input SLR data is available at the open access EUROLAS Data Center (EDC).

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
