# Peer review of "Analysis of the Quality of SLR Station Coordinates Determined from Laser Ranging to the LARES Satellite"

_sensors, 2021, doi:10.3390/s21030737_

Round 1

Reviewer 1 Report

# Report for *Analysis of quality of SLR station coordinates determined from laser ranging to the LARES satellite* by Schillak, Lejba, Michałek

The article describes an interesting work on an important issue, related to the inclusion of LARES laser ranging data in the analysis procedure used to generate the ITRF frame. As such, the topic deserves major attention. However, before being considered for publication, several parts in the article should be improved. It is not clear whether a truly global (e.g. normal equations inversion) analysis has been performed. The very tight adopted editing strategy leads to a low number of considered stations, with possible serious biases in the global reconstruction of their positions. The authors should strive to provide more evidence for the effectiveness of their procedure (and of how it integrates with the ILRS Analysis Centers), or at least improve the exposure clarity.

---

Specific comments (for each comment the corresponding line numbers are indicated):

+ 19-21 The sentence is not clear at all. Please explain its meaning and rephrase it.

+ 29-30 Frame dragging should exists regardless of the primary moment of inertia value (provided it is non zero). Of course, for the effect being detectable, the moment of inertia should be large enough.

+ 35-36 The sentence is not clear. In any case, the inclusion of LARES data into the ITRF generation procedure is, as far as I know, well under study by the geodetic community.

+ 82 It is not clear the meaning of "Materials". Please explain.

+ 94 (tab 2) Please explain the difference between "Earth tides" and "Earth tide model".

+ 94 (tab 2) The orbit integration reference system has been selected as the one "defined at 0.0 h of the first day of each arc". Please explain the rationale for this choice, compared e.g. to selecting J2000.

+ 94 (tab 2) Has post-seismic deformation model been included into SLRF2014 implementation?

+ 94 (tab 2) From the number (14) of used acceleration parameters, it seems that they are applied two times per day. Please confirm this. Adding so large a number of parameters surely affects the solution. Please comment on this.

+ 94 (tab 2) Is the wavelength value of 532 nm valid for all stations?

+ 103 How the time-dependent part of the geopotential has been modeled?

+ 104 The authors say that "The influence of atmospheric drag was not found". This seems in contradiction with what found, e.g., in Pardini et al (2017) (reference 5 in the article), in which a well-defined LARES orbit semimajor axis decay (caused for the most part by atmospheric drag) has been measured. One could think that this effect has been absorbed by the large number of acceleration parameters added. Please clarify.

+ 110-112 It is not clear the reason the Authors did not include their estimate of range biases, while they at the same time state that it is important for the analysis. Please clarify.

+ 113-115 One could think that the conditions imposed for the iteration convergence are too restrictive, especially in view of the relatively large number of rejected arcs. Please discuss this issue carefully.

+ 135-137 The meaning of the stated transformation is not clear. Do the Authors mean "geodetic" instead of "topocentric"? And what is the purpose of this? Please explain.

+ 135-137 From what stated in the text, it seems that the Authors used a plain multi-arc technique, without a global solution (e.g. normal equations inversion). Is that true? Please explain.

+ 144-148 The authors state that "Higher quality of results for European stations is clearly visible", and point to "a large number of stations in a small area". This, on one hand, is clear, but on the other hand could point to a serious bias in the analysis. Indeed, the precise orbit determination is in any case global, and restricting to 17 stations only is surely of no help. I have the feeling that in some way the orbit is "pulled" towards the European part by the estimation procedure. Please carefully comment on this.

+ 153-154 This sentence seems to contradict the outcome of the analysis (144-148). Please comment on this.

+ 158-159 It seems that the long-term stability has been defined as the scatter (error bars in tab 5) of range bias. If so, please motivate this choice.

+ 228-230 The quality improvement could result also from an improvement of the global geometry, not only from an increase in the number of normal points.

+ 258-260 Please explain this sentence, which furthermore seems in contradiction to what stated in 153-154.

+ 290 What stated by the authors on a possible significant reduction of Earth albedo effect in the case of LARES 2 is at least questionable. Indeed, LARES 2 is planned to have an orbit similar to that of LAGEOS, for which Earth albedo effects (though not huge) are known to be significant.

---

Editorial comments (for each comment the corresponding line numbers are indicated):

+ 9 satellite (LAser RElativity Satellite) --> (LAser RElativity Satellite) satellite

+ 17 program GSFC NASA GEODYN-II --> NASA GSFC GEODYN-II software

+ 18 is the --> is nearly the

+ 22 uncertainty --> uncertainty of

+ 23 than only for LAGEOS satellites --> than the LAGEOS-only one

+ 25 law satellite orbits --> low-altitude satellite orbits

+ 28 satellite (LAser RElativity Satellite) --> (LAser RElativity Satellite) satellite

+ 30 an inertial system --> a locally defined inertial system

+ 30 frame-dragging --> frame dragging

+ 31 papers eg [4-6] --> papers, e.g. [4-6]

+ 33 law geodesy satellites in determination --> low-altitude geodesy satellites in determining

+ 34 discuss from --> discussed since

+ 42 quality results --> the quality of results

+ 43 is this solution can --> can this solution be

+ 44 low geodetic satellite to co-create --> low-altitude geodetic satellite to co-establish

+ 54 too large diameter causing an increase in --> a too large diameter causing significant

+ 59 The LARES satellite built by the Italian Space Agency (ASI) was --> The LARES satellite, built by the Italian Space Agency (ASI), was

+ 69 in the satellite --> in satellite

+ 71 non-gravity --> non-gravitational

+ 72 non-gravity --> non-gravitational

+ 73 A/M = 2.69 x10 -4 m2/ kg --> A/M = 2.69 x 10-4 m^2/kg

+ 74 15.3 g/ cm3 --> 15.3 g/cm^3

+ 84 GSFC NASA GEODYN-II orbital program --> NASA GSFC GEODYN-II orbital software

+ 84 the results of laser ranging --> the laser ranging data

+ 88 The average RMS of normal points from determined --> The average RMS of determined

+ 90 was the same --> was nearly the same

+ 91 program --> software

+ 94 (tab 2) Third body --> Third-body

+ 94 (tab 2) MSIS86 --> MSIS-86

+ 94 (tab 2) 3,986004415 --> 3.986004415

+ 94 (tab 2) model Love model --> Love model

+ 94 (tab 2) model Mendes-Pavlis --> Mendes-Pavlis model

+ 94 (tab 2) Residua --> Residual

+ 94 (tab 2) 3RMS --> 3DRMS

+ 105 components --> components applied

+ 125 base --> basis

+ 126 was --> were

+ 135 GSFC NASA GEODYN-II program --> NASA GSFC GEODYN-II software

+ 151 program --> software (two occurrences)

+ 154 for three --> for the three

+ 155 on --> in

+ 159 long term --> long-term

+ 160 long term --> long-term

+ 184 Long term --> Long-term

+ 198 is determination --> is the determination

+ 205 a --> the

+ 205 were --> are

+ 206 was --> is

+ 208 Please add a ";" after the equation.

+ 209 components North, East and Up are computed analogously (instead X, Y, Z) and final --> the components North, East and Up are computed analogously (instead of X, Y, Z) and the final

+ 211 and three solutions are presented --> and the three solutions is presented

+ 216 further researches --> further research and

+ 220 for three --> for the three

+ 223 determining is define --> determination is defined

+ 225 of determined --> of the determined

+ 227 without Svietloe --> except Svetloe

+ 229 determining position of the station --> determined station positions

+ 252 5 --> 10

+ 258 to be much --> being much

+ 263 is the same --> is nearly the same

+ 266 satellite changed --> satellite is in the range

+ 273 satellite change --> satellite is in the range

+ 274 change --> is

+ 274-275 is the best result from --> is (best result) from

+ 284 coordinates of the station --> station coordinates

+ 287 (ITRF) --> (ITRF) release

+ 288 LARES-2 --> LARES 2

+ 291 should be --> should

+ 292 satellite, and in the total solution due --> satellites, and in the overall solution, due

+ 293 determining the coordinates of the stations --> stations coordinates determination

+ 294-295 only from two LAGEOS satellites --> the LAGEOS satellites-only one

+ 296 need --> needs

+ 309 Reference 1. apparently is not cite in the text; please check.

+ 374 geographic --> geodetic

Reviewer 2 Report

The authors demonstrate the results of station coordinate determination using laser ranging data of LARES satellite. The quality assessment or their determination by comparison with the results from LAGEOS satellites were also carried out. Finally, the authors present that the solution using LARES satellite is slightly better than only using the LAGEOS satellite. The main focus of the research is clear, and the results support the conclusion well. The contribution to the geodetic research field of this paper is important. ITRF development is a fundamental study for various satellite-based applications; therefore, laser-ranging data from the LARES mission can improve products based on ITRF information. From this perspective, this paper's approach and results are appropriate and useful for publication in Sensors. I did not find the critical concern of this manuscript. I made some suggestions and minor comments for the publication of Sensors.

Reviewer 3 Report

Dear Authors,

please see a few of my comments in the attached copy of the manuscript.

I do not have any major objections to your work.

Congratulations!

Round 2

Reviewer 1 Report

After this review round I think the article is worth and ready for publication.